# Furosemide-Induced Nephrocalcinosis in Premature Neonates: A Critical Review of Observational Data

**DOI:** 10.3390/children12111442

**Published:** 2025-10-24

**Authors:** John Dotis, Alexandra Skarlatou, Maria Fourikou, Athina Papadopoulou, Elpis Chochliourou

**Affiliations:** 1Third Department of Pediatrics, Aristotle University of Thessaloniki, Hippokration Hospital, 54642 Thessaloniki, Greece; mfour@auth.gr (M.F.); echochl@auth.gr (E.C.); 2Neonatal Intensive Care Unit, Pediatric Department, Democritus University of Thrace, University General Hospital of Alexandroupolis, 68100 Alexandroupolis, Greece; alskarlatou@pgna.gr; 3First Department of Pediatrics, Aristotle University of Thessaloniki, Hippokration Hospital, 54642 Thessaloniki, Greece; athinapap80@yahoo.gr

**Keywords:** furosemide, nephrocalcinosis, preterm neonates, dose-dependent risk, renal outcomes

## Abstract

**Highlights:**

**What are the main findings?**
•Furosemide is linked to a dose-related risk of nephrocalcinosis in preterm infants.•Although the condition resolves with discontinuation of medication in most cases, it can persist especially with prolonged exposure.

**What is the implication of the main finding?**
•Cautious dosing and close monitoring are crucial to minimizing renal complications.•Further research is needed to establish optimal dosing and long-term safety.

**Abstract:**

Background/Objectives: Furosemide is frequently used in preterm neonates for respiratory and fluid management but has been linked to nephrocalcinosis (NC), a renal complication with unclear long-term impact. Clarifying this association is crucial for safe diuretic use. Methods: A focused literature review included observational studies published between 1982 and 2025 reporting NC incidence by renal ultrasound in preterm infants receiving furosemide. Data on sample size, gestational age, birth weight, NC prevalence, and furosemide dosing/duration were extracted. Results were synthesized descriptively. Results: Twenty-two studies with 1489 infants were included. NC prevalence ranged 6–83%, higher in infants <32 weeks’ gestation or <1500 g. Across studies, incidence clustered at 17–41% between 4 weeks and term-equivalent age. Cumulative furosemide doses were generally three- to fourfold higher in NC groups (10–19 mg/kg cumulative vs. ≤5 mg/kg cumulative, *p* < 0.001). A dose-dependent risk was noted, with odds ratios increasing above a cumulative dose of 10 mg/kg. Some studies found no significant dose–response, indicating variability and confounding factors. NC was detected during NICU stay or around term-equivalent age; ~60% resolved after discontinuation, while persistent cases were associated with prolonged exposure and renal dysfunction. A recent multicenter, dose-escalation randomized trial showed that carefully dosed furosemide (≤2 mg/kg/day for 28 days) did not increase NC risk, though electrolyte disturbances were more frequent. Conclusions: Evidence supports a dose-related association between furosemide and NC in preterm infants. When administered cautiously within defined limits, risk may be mitigated. Careful dosing, monitoring, and further studies are essential for safe use.

## 1. Introduction

Furosemide is one of the most commonly prescribed loop diuretics in neonatal intensive care units (NICUs), mainly used for the management of pulmonary edema, fluid overload, and bronchopulmonary dysplasia in preterm infants. Its pharmacological action, inhibition of sodium and chloride reabsorption in the thick ascending limb of Henle’s loop, leads to enhanced urinary calcium excretion [1]. While clinically effective in improving respiratory and fluid status, this mechanism has been consistently linked to the development of nephrocalcinosis (NC) [1,2].

In premature neonates, the occurrence of NC is well documented, with prevalence rates ranging from 6% to over 60%, depending on gestational age, birth weight, cumulative drug exposure, and variations in study design [3]. Evidence from observational cohorts suggests that higher cumulative doses of furosemide, particularly above 10 mg/kg/day, substantially increase the risk. However, not all studies confirm a strict dose–response relationship, indicating that additional neonatal factors such as extreme prematurity, prolonged parenteral nutrition, or co-administered therapies, likely contribute [4,5].

The natural history of NC in preterm infants is heterogeneous. In many cases, calcifications regress spontaneously after discontinuation of furosemide, suggesting a transient and reversible phenomenon. Nevertheless, NC has persisted in a notable proportion of infants; in certain cohorts, it has been associated with impaired renal growth, tubular dysfunction, decreased glomerular filtration, and the development of early-onset hypertension [6]. These observations raise the possibility that NC may not always be a benign imaging finding but rather an early marker of vulnerability in renal development.

## 2. Materials and Methods

### 2.1. Study Design and Protocol Registration

This review was conducted as a systematic synthesis of observational studies, including cohort studies from previous years and case–control studies, as well as clinical trials from the last 5 years. These studies evaluated the association between furosemide exposure and NC in premature neonates (gestational age <37 weeks). The methodology adhered to the Preferred Reporting Items for Systematic reviews and Meta-Analyses (PRISMA) 2020 reporting standards to ensure rigor and transparency [7]. A protocol was developed a priori and registered in PROSPERO (Registration number 2025 CRD420251146261. Available online at https://www.crd.york.ac.uk/PROSPERO/view/CRD420251146261 (accessed on 13 September 2025) to enhance methodological consistency and reduce potential bias.

### 2.2. Study Eligibility Criteria

Inclusion criteria:•Studies on premature infants who received at least one dose of furosemide in the NICU.•The studies reported nephrocalcinosis (NC) confirmed by ultrasound or similar imaging.•The studies had a comparator group of premature infants without furosemide exposure (when available).

Exclusion criteria:•Animal studies, narrative reviews, and editorials.•Conference abstracts without full data.•Studies that did not assess NC as an outcome.

### 2.3. Information Sources and Search Strategy

A systematic search was performed in MEDLINE (via PubMed), EMBASE, and Google Scholar from database inception through 1 May 2025. The search strategy combined controlled vocabulary (MeSH/Emtree) as well as free-text terms relating to the exposure, population, and outcome of interest, for example: (“furosemide” [MeSH] OR “furosemide”) AND (“premature infant” [MeSH] OR “preterm infant” OR “neonate”) AND (“nephrocalcinosis” OR “renal calcification”). Reference lists of included studies and relevant reviews were manually screened to capture additional eligible publications.

### 2.4. Study Selection and Data Extraction

All records were imported into reference management software and duplicates were removed. Two reviewers independently screened titles and abstracts, with potentially relevant studies undergoing full-text review. Disagreements were resolved through discussion with a third reviewer. A standardized extraction form was used to record study-level data, including publication year, country, design, sample size, gestational age and birth weight, details of furosemide administration (dose, route, duration), comparator group characteristics (if present), diagnostic method and timing of NC assessment, follow-up duration, and reported outcomes. Information on funding sources and conflicts of interest was also collected where available.

### 2.5. Risk of Bias and Quality Assessment

Randomized controlled trials (RCTs) were evaluated using the Cochrane Risk of Bias tool, which examines sequence generation, allocation concealment, blinding, completeness of outcome data, and selective reporting [8]. Non-randomized studies were appraised with the ROBINS-I instrument, assessing confounding, participant selection, misclassification, deviations from intended interventions, missing data, and outcome measurement [9]. Each study was graded as having low, moderate, serious, or critical risk of bias. The overall strength of the body of evidence was assessed following guidance from the U.S. Agency for Healthcare Research and Quality [10].

### 2.6. Data Synthesis

Extracted data were summarized in comparative tables to highlight study characteristics and findings. Descriptive statistics were applied, with continuous variables presented as means or medians (with ranges or interquartile ranges) and categorical variables as frequencies or percentages. Where feasible, measures of association such as odds ratios (ORs) or risk ratios (RRs), with 95% confidence intervals (CIs), were reported. Given the substantial heterogeneity in study designs, populations, definitions of NC, dosing regimens, and imaging protocols, quantitative synthesis was not performed, and findings were synthesized narratively.

### 2.7. Ethical Considerations

As this study relied exclusively on previously published data, no new patient recruitment or collection of individual-level information was performed. Institutional review board approval as well as informed consent were therefore not required.

### 2.8. Statistical Analysis

All extracted data were entered into a master database using Microsoft Excel. GraphPad Instat 3.10 (GraphPad Inc., San Diego, CA, USA) was used to generate descriptive statistics for continuous and categorical variables. A two-sided *p*-value of <0.05 was considered statistically significant. Results were synthesized narratively with emphasis on the predefined outcomes. No formal meta-analysis was undertaken due to heterogeneity across studies.

## 3. Results

### 3.1. Studies Identified Through the Search Strategy

From an initial 2053 records identified through PubMed, EMBASE, and Google Scholar, the systematic review process removed 1621 duplicates and 3 additional irrelevant records, yielding 429 articles for screening. After reviewing titles and abstracts, 248 studies were excluded (e.g., case reports, reviews, or non-human research). The remaining 182 full-text articles were then assessed for eligibility, with 160 being excluded for not meeting the inclusion criteria for participants, outcomes, or safety data. Ultimately, this process resulted in the inclusion of 22 studies, with a combined sample size of 1489 infants (Figure 1). This final cohort consisted of 1 multicenter RCT, along with 21 observational reports (20 cohort studies and 1 retrospective case–control study) [2,11,12,13,14,15,16,17,18,19,20,21,22,23,24,25,26,27,28,29,30,31].

### 3.2. Findings on Furosemide and NC

Results from the observational studies were mixed, with 12 reporting a significant, often dose-dependent association between furosemide exposure and NC, while 10 studies found no such relationship. The case–control study identified furosemide exposure and impaired postnatal growth as independent risk factors for NC [31]. Most studies were single center, with two exceptions, one including infants from 2 centers [21] and the other from 17 centers [2]. Three studies resembled case series due to the lack of appropriate control groups; one reported a positive association with furosemide, while two did not [11,19,23]. The multicenter RCT provided higher-quality evidence, showing no increase in overall NC incidence but a higher rate of electrolyte abnormalities [2].

Table 1 provides an overview of all included studies, summarizing reference number, year, country, study design, population, sample size, primary outcomes, and key findings [2,11,12,13,14,15,16,17,18,19,20,21,22,23,24,25,26,27,28,29,30,31].

There was notable variability in inclusion criteria, timing of renal ultrasonography (RUS), and follow-up duration. Across these studies, all pre-term infants demonstrated sonographic evidence of NC prior to NICU discharge. In studies that included post-discharge imaging, NC resolved in 44–100% of infants over time [12,14,17,19,20,23,27,28,30,31]. Long-term outcome data remain incomplete, as information was unavailable for infants with severe illness who died during the NICU stay or those lost to follow-up. Differences in furosemide exposure, cumulative dose together with duration of treatment likely contributed to the observed variability in the persistence and resolution of NC.

### 3.3. Overall Prevalence and Dosing

Across the included studies, the prevalence of NC ranged from 6% to 83%, with higher rates observed in infants born at <32 weeks’ gestation or with birth weight <1500 g. Incidence clustered between 17% and 41% from 4 weeks of life to term-equivalent age. The majority of studies reported that cumulative furosemide doses were substantially higher in infants who developed NC (10–19 mg/kg/day vs. ≤5 mg/kg/day, *p* < 0.001) and several studies demonstrated a dose-dependent increase in risk, particularly when cumulative doses exceeded 10 mg/kg. In most reports, nephrocalcinosis was associated with furosemide courses lasting longer than 2–3 weeks or cumulative treatment exceeding 10–15 mg/kg. However, four studies found no significant dose–response relationship, indicating the influence of confounding factors such as severity of illness, co-administered medications, and gestational age [14,15,19,22].

During NICU admission or at term-equivalent age, NC was detected most often. Approximately 64% of cases resolved spontaneously after discontinuation of furosemide, whereas persistent NC was associated with prolonged exposure, higher cumulative doses, and evidence of renal dysfunction (e.g., reduced creatinine clearance or tubular impairment) [12,14,17,19,20,23,27,28,30,31]. The only multicenter RCT included [2] reported that carefully dosed furosemide (≤2 mg/kg/day for up to 28 days) did not increase NC incidence, although electrolyte disturbances were more frequent.

### 3.4. Resolution and Risk of Bias

Risk-of-bias assessment revealed that older and most observational studies had serious to moderate risk, particularly in confounding domains (Table 2), largely due to lack of control groups, insufficient adjustment for illness severity, or small sample sizes. Studies with robust methodology and multivariate analyses [15,20,27] consistently demonstrated a relationship between higher furosemide exposure and NC development.

## 4. Discussion

Current evidence does not conclusively demonstrate risk of NC with furosemide exposure, with overall quality of the available evidence remaining low. Nevertheless, many observational studies suggest a heightened risk of NC in infants receiving furosemide, particularly at higher cumulative doses. This pattern is most apparent in older studies, which were often limited by small sample sizes, heterogeneity in study design, variable follow-up, as well as inadequate control for confounders [11,13,20,27]. NC occurs most frequently in the most immature infants, especially those born before 32 weeks’ gestation or weighing less than 1500 g at birth. In these studies, infants who developed NC typically received cumulative furosemide doses two- to fourfold higher than those without NC, indicating a dose-dependent effect. Persistent NC was linked to prolonged drug exposure, impaired tubular function, and reduced creatinine clearance, whereas most cases resolved after discontinuation [16,25].

Ιn contrast, more recent evidence is clearer: a large retrospective case–control study identified furosemide exposure and postnatal growth restriction as independent risk factors for NC [31], while a multicenter RCT showed that carefully titrated furosemide did not increase overall NC incidence, though electrolyte disturbances remained more frequent [2]. Despite these findings, differences in follow-up schedules, imaging protocols, and NC definitions continue to limit the ability to draw definitive conclusions [21,30,31]. Most included studies were observational, which limits causal inference. Confounding factors such as illness severity, duration of mechanical ventilation, co-administered medications, and underlying comorbidities were often inadequately controlled, contributing to variability in outcomes [13,15,20,23,28]. Additionally, differences in populations, dosing regimens, follow-up, and imaging frequency further hinder comparability and preclude meta-analytic synthesis [24,29].

Despite limited evidence for long-term efficacy in improving outcomes such as bronchopulmonary dysplasia, furosemide is frequently administered to preterm infants to reduce fluid overload and facilitate weaning from respiratory support [28,29]. Considerable variability in NICU prescribing practices reflects the absence of standardized guidelines, with some centers limiting exposure and others administering the drug more liberally [23,30]. Nevertheless, furosemide remains widely used in neonatal practice.

Given the potential for NC and related renal complications, clinicians should carefully consider individual risk factors such as gestational age, birth weight, cumulative furosemide exposure, in addition to postnatal growth when initiating therapy. Monitoring protocols should include serial RUS, periodic assessment of renal function through serum creatinine and cystatin C, urinalysis for proteinuria, as well as electrolyte surveillance. Infants who develop persistent NC may require extended follow-up, since they could be at heightened risk for chronic kidney disease in later life [17,31].

This systematic review has several limitations. Only 1 multicenter RCT was available, while the remaining 21 studies were observational, including cohort and 1 case–control study, which inherently limits the ability to establish causal relationships. The observational studies varied considerably in sample size, study design, inclusion criteria, timing of RUS, and follow-up duration, further restricting comparability. Many of these studies were conducted decades ago, prior to widespread use of antenatal steroids, surfactant therapy, together with non-invasive ventilation, which have substantially improved survival and morbidity outcomes in preterm infants. As a result, findings from older cohorts may not fully reflect current neonatal care practices. The search was limited to English-language publications, potentially omitting relevant studies and differences in terminology could have influenced study identification. Finally, the heterogeneity in outcome definitions, population characteristics as well as duration of follow-up precluded a quantitative meta-analysis, limiting the ability to generate pooled estimates of risk. Specifically, the heterogeneity in study design, dosing definitions (daily vs. cumulative exposure), population characteristics, and outcome reporting precluded the possibility of performing a quantitative meta-analysis to estimate the effect of cumulative furosemide dose on the development of NC. Moreover, most studies did not report detailed laboratory data such as spot urine calcium-to-creatinine ratio, presence of hypercalcemia, or deranged renal function tests and electrolytes, precluding any definitive conclusions on the association of these parameters with NC. Nevertheless, this review represents an important effort to clarify the association between furosemide exposure and NC in preterm infants.

Future research must prioritize rigorously designed multicenter RCTs, such as the last study included in this review [2], to establish the safety and optimal dosing of furosemide in this vulnerable population. These investigations should employ standardized definitions of NC, consistent imaging schedules, and long-term renal outcome assessments. Dose-escalation approaches may help determine the threshold at which furosemide contributes to NC while minimizing unnecessary exposure. Implementing such comprehensive strategies will generate stronger, evidence-based guidance for clinicians, enabling careful balancing of furosemide’s therapeutic benefits against potential nephrotoxic and long-term renal risks in this high-risk population.

## 5. Conclusions

Furosemide remains a widely used therapeutic agent in preterm infants. While high-quality evidence from a multicenter RCT shows that carefully dosed furosemide does not increase the overall incidence of NC, data from numerous observational studies suggest a dose-dependent risk, especially in the most immature infants and those receiving higher cumulative doses. Therefore, careful dose management, close monitoring, together with consideration of individual risk factors are essential to minimize potential renal complications. To address this critical gap in the literature, future research must prioritize rigorous, multicenter trials to elucidate optimal furosemide dosing regimens and establish long-term safety profiles for this vulnerable cohort.

## Figures and Tables

**Figure 1 children-12-01442-f001:**
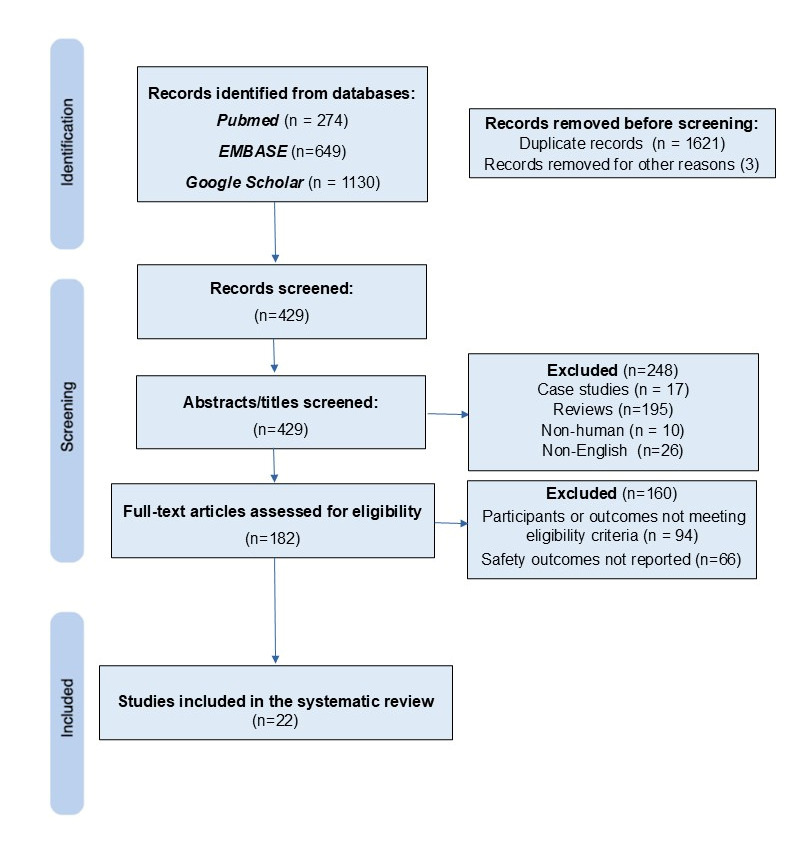
PRISMA flow diagram showing the number of studies screened, included, and excluded.

**Table 1 children-12-01442-t001:** Summary of studies examining risk of nephrocalcinosis in premature infants.

Ref. No (Year)	Country	Study Design	Population and Sample Size	Primary Outcome	Key Findings
[11] (1982)	USA	Cohort	10 premature infants with NC	RUS during NICU admission	All infants were exposed to furosemide for at least 12 days at a dose of at least 2 mg/kg/day before their NC diagnosis.
[12] (1988)	USA	Cohort	36 infants with BW ≤ 1500 g	RUS at 12 months of age	NC occurred in 3 of 32 infants (9%) receiving chronic furosemide therapy and resolved in 67% of cases.
[13] (1988)	USA	Cohort	31 infants with BW < 1500 g	RUS in the third week of life and every three weeks thereafter	NC was diagnosed in 64% of infants. Furosemide exposure was significantly more common in the NC group (65% vs. 9%; *p* < 0.001).
[14] (1988)	USA	Cohort	17 premature infants with NC treated with furosemide; 3 premature infants treated with furosemide without NC (control group)	RUS during NICU admission	There was no difference in the average daily dose or duration of furosemide between the groups. Resolution of NC observed in 44% of cases.
[15] (1991)	USA	Cohort	79 infants with GA < 32 weeks	Serial RUS	NC was diagnosed in 27% of infants. No significant difference was found in the mean total dose of furosemide.
[16] (1991)	USA	Cohort	117 infants with BW < 1750 g and BPD treated with furosemide	RUS prior to discharge and at 3–6 month intervals	17% of infants showed evidence of NC before discharge. Infants who remained on furosemide were more likely to have persistent NC compared to those who stopped the medication (*p* < 0.001).
[17] (1992)	USA	Cohort	27 infants with BW < 1500 g divided into 3 groups	RUS and laboratory testing for glomerular and tubular kidney function	Infants with NC had lower creatinine clearance and higher tubular dysfunction compared to the other groups. Resolution observed in 60% of cases.
[18] (1992)	USA	Cohort	11 premature infants with post-hemorrhagic hydrocephalus treated with furosemide and acetazolamide	Serial RUS	45% of infants showed evidence of NC. No correlation was found between treatment duration, total dosage, and the development of renal calculi.
[19] (1996)	USA	Cohort	13 premature infants with NC exposed to furosemide, divided into 2 groups: resolution of NC and persistent NC	Serial RUS	There was no difference in the duration or cumulative dose of furosemide between the groups. Resolution observed in 46% of cases
[20] (1999)	Finland	Cohort	129 infants with BW < 1500 g	RUS at 2, 6, and 12 weeks of age	20% of infants were diagnosed with NC. The mean cumulative doses of furosemide were significantly higher in infants with NC. Resolution observed in 84% of cases.
[21] (2000)	The Netherlands	Cohort	215 infants with GA < 32 weeks	RUS at 4 weeks of life and at term	NC was diagnosed in 33% of infants at 4 weeks and 41% at term. At term, furosemide exposure was higher in the group with NC (32% vs. 18%).
[22] (2001)	USA	Cohort	101 infants with GA < 32 weeks or BW < 1500 g	RUS at 1 month of age and at term or NICU discharge	16% of infants were diagnosed with NC. The median total dose of furosemide was not significantly different between the groups.
[23] (2002)	Germany	Cohort	16 infants with GA < 37 weeks with an NC diagnosis	RUS during NICU admission and every 3–6 months following discharge	NC persisted in 33% of infants who received follow-up. Infants whose NC resolved received lower dosages of furosemide. Resolution observed in 69% of cases.
[24] (2004)	Germany	Cohort	114 infants with BW < 1500 g, divided into 2 groups: (1) with NC; (2) without NC	RUS every 2 weeks during NICU admission	There was no difference in the duration of furosemide therapy between the two groups.
[25] (2004)	Thailand	Cohort	36 infants with GA < 32 weeks and BW < 1250 g	RUS prior to NICU discharge	39% of infants were diagnosed with NC. The mean cumulative dose and mean duration of furosemide were significantly higher in infants with NC.
[26] (2004)	USA	Cohort	33 infants with GA < 28 weeks and BW < 990 g, δexamethasone trial group	RUS on study entry, day of life 28, and at discharge or 36 weeks postmenstrual age	83% of infants with complete data were diagnosed with NC. Furosemide was used infrequently, and 88% of infants who never received it still developed NC.
[27] (2010)	Germany	Cohort	55 infants with GA < 32 weeks and BW < 1500 g	RUS obtained after the first month of life	27% of infants were diagnosed with NC. The strongest independent risk factor was furosemide therapy with a cumulative dose over 10 mg/kg. Resolution observed in 100% of cases during 2–3 years of follow-up.
[28] (2011)	USA	Cohort	102 infants with GA < 34 weeks and BW < 1500 g	RUS at term or prior to NICU discharge	6% of infants were diagnosed with NC. Furosemide exposure was more common in the NC group (33% vs. 3%). Resolution observed in 50% of cases.
[29] (2014)	North Korea	Cohort	52 infants with BW < 1500 g	RUS at 4 and 8 weeks of life	Furosemide exposure did not significantly differ between the groups with and without NC.
[30] (2014)	Saudi Arabia	Cohort	97 infants with GA ≤ 34 weeks	RUS at the first week of life, at term, and at one year corrected age	Furosemide exposure was more common in the NC group (50% vs. 16%). Resolution observed in 57% of cases.
[31] (2022)	USA	Retrospective case–control study	265 infants with GA < 32 + 6 weeks and ≤1500 g, 80 infants with NC	RUS at 35 weeks corrected GA	Furosemide exposure and postnatal growth were identified as independent risk factors for NC. Resolution observed in 61% of cases.
[2] (2025)	USA	Randomized clinical trial	80 infants with GA < 29 weeks	RUS prior to NICU discharge	Furosemide did not increase the overall incidence of adverse events, including NC.

NC, nephrocalcinosis; RUS, renal ultrasound; NICU, neonatal intensive care unit; BW; birth weight; GA, gestational age.

**Table 2 children-12-01442-t002:** Quality assessment of studies examining risk of nephrocalcinosis in premature infants.

Ref. No (Year)	Risk of Bias	Main Limitation/Strength
[11] (1982)	Critical	*Limitation**:* Critical risk of bias due to lack of control group and no statistical analysis of NC-furosemide association
[12] (1988)	Serious	*Limitation:* No statistical tests were performed to assess the association of NC and furosemide
[13] (1988)	Serious	*Limitation:* The study did not control for illness severity, despite lower birth weight and gestational age being associated with the outcome and furosemide exposure
[14] (1988)	Serious	*Limitation:* Lack of adjustment for illness severity and a small control group
[15] (1991)	Moderate	*Strength:* Controlled for confounding with multivariate analyses and assessed dose–response relationship
[16] (1991)	Moderate	*Strength:* High follow-up rate and all infants had chronic lung disease, allowing consistent outcome assessment
[17] (1992)	Moderate	*Strength:* Robust comparator groups and included long-term follow-up
[18] (1992)	Serious	*Limitation:* No statistical tests were performed to assess the NC-furosemide association. The study also failed to report the frequency of NC in unexposed infants
[19] (1996)	Moderate	*Strength:* Study groups had similar illness severity and provided long-term follow-up with serial ultrasounds while also evaluating a dose–response relationship
[20] (1999)	Moderate	*Strength:* Dose–response relationship
[21] (2000)	Moderate	*Strength:* Large sample size with control group without NC
[22] (2001)	Moderate	*Strength:* Multivariate analyses were used to control for other risk factors for NC, and a dose–response relationship was also evaluated
[23] (2002)	Serious	*Limitation:* Absence of a control group without NC
[24] (2004)	Moderate	*Strength*: Large sample size and appropriate control groups
[25] (2004)	Moderate	*Strength*: Appropriate control group included; dose–response evaluated
[26] (2004)	Moderate	*Strength*: Comparable illness severity among all infants
[27] (2010)	Moderate	*Strength*: Multivariate analysis with dose–response assessment
[28] (2011)	Serious	*Limitation*: Lack of adjustment for illness severity and a low incidence of NC in the sample.
[29] (2014)	Serious	*Limitation*: No adjustment was made for the severity of illness
[30] (2014)	Serious	*Limitation*: No adjustment was made for the severity of illness
[31] (2022)	Serious	*Limitation*: The case and control groups differed in terms of morbidity, which introduces a high risk of confounding
[2] (2025)	Low	*Strength*: Multicenter randomized controlled trial; minimizes bias and systematically assessed NC

NC, nephrocalcinosis.

## Data Availability

The data supporting the reported results are from previously published studies. A list of all included studies is provided in the manuscript’s reference section. The original data presented in the study are openly available in all the academic databases mentioned in the method section.

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
