# Peer review of "Furosemide-Induced Nephrocalcinosis in Premature Neonates: A Critical Review of Observational Data"

_children, 2025, doi:10.3390/children12111442_

Round 1

Reviewer 1 Report

Comments and Suggestions for Authors

In the highlights section-sentence 13- can be re-structured as ' although the condition resolves with discontinuation of medication in most cases, it can persist especially with prolonged exposure.  In the discussion section- sentence 217-re-phrase as 'Current evidence does not conclusively demonstrate risk of NC with furosemide exposure with overall quality....'.  The results suggest association of NC with prolonged exposure- based on the review of literature is there a documented length ( number of days)? 

• What is the main question addressed by the research? This is a review article on risk of furosemide use and nephrocalcinosis in premature neonates. 
• Do you consider the topic original or relevant to the field? Does it
address a specific gap in the field? Please also explain why this is/ is not
the case. As the current literature is inconclusive, the review article is a good compilation of relevant studies and results to give future directions.
• What does it add to the subject area compared with other published
material? review articles compiling all the literature adds to current knowledge of the problem
• What specific improvements should the authors consider regarding the
methodology? This is a review article and the authors have included selection of data, inclusion/exclusion with PRISMA flow chart.
• Are the conclusions consistent with the evidence and arguments presented
and do they address the main question posed? Please also explain why this
is/is not the case. The conclusions are based on literature review with attention to more recent publications
• Are the references appropriate? yes
• Any additional comments on the tables and figures tables are appropriate

Author Response

We would like to sincerely thank Reviewer 1 for the thoughtful comments and helpful suggestions, which have improved the quality and clarity of our manuscript. We have carefully revised the text as indicated below.

Comment 1. In the highlights section—sentence 13—can be re-structured as: “Although the condition resolves with discontinuation of medication in most cases, it can persist especially with prolonged exposure.”

Response: We appreciate this helpful suggestion. The sentence in the Highlights section has been rephrased exactly as recommended to improve clarity and flow (revised ms, lines 13,14)

Comment 2. In the discussion section—sentence 217—rephrase as: “Current evidence does not conclusively demonstrate risk of NC with furosemide exposure with overall quality….”

Response: We thank the reviewer for this stylistic improvement. The sentence has been revised to read as follows: “Current evidence does not conclusively demonstrate risk of NC with furosemide exposure with overall quality of the available evidence remains low.” (revised ms, lines 235,236). This rephrasing enhances precision and academic tone.

Comment 3: The results suggest association of NC with prolonged exposure—based on the review of literature is there a documented length (number of days)?

Response: We thank the reviewer for this valuable observation. We have now included specific information on treatment duration in the Results section (revised ms, lines 211-213). Based on the reviewed studies, NC was most often reported after furosemide courses exceeding 2–3 weeks or cumulative doses greater than 10–15 mg/kg. This addition clarifies the relationship between the duration of exposure and the risk of NC, supported by data from multiple observational studies.

Reviewer 2 Report

Comments and Suggestions for Authors

The manuscript titled "Furosemide-Induced Nephrocalcinosis in Premature Neonates: A Critical Review of Observational Data" is well-organized. The objectives and background are clearly articulated, providing valuable information. However, several points require attention before the article can be accepted:

  1. In the manuscript, the abbreviation NC is used for nephrocalcinosis. However, in Table 1, there are instances of NC/NL in the key findings of reference 16. Please provide the full form of the abbreviation NC/NL in the manuscript.
  2. In Line 183, the authors describe that “NC resolved in 44–100% of infants over time.” However, Table 1 does not include information regarding post-discharge imaging. The authors should either add this evidence to Table 1 or provide a new table to summarize it.
  3. In the section on "Overall Prevalence and Dosing," the authors mention that “the majority of studies reported cumulative furosemide doses were substantially higher in infants who developed NC (10–19 mg/kg/day vs ≤5 mg/kg/day, p<0.001).” If there are more than five studies included, the authors may consider performing a meta-analysis to verify the effect of cumulative furosemide dose on the development of NC.
  4. In Line 195, the authors should indicate the number of cited references that reported no significant dose–response relationship.
  5. In Line 199, the authors should indicate the number of cited references that reported approximately 60% of cases resolved spontaneously after discontinuation of furosemide.
  6. In the discussion section, the authors may consider suggesting monitoring protocols, such as the frequency of serial renal ultrasounds (during admission and post-discharge) and the frequency of blood tests for electrolytes and renal function.

Author Response

We would like to thank Reviewer 2 for the constructive feedback and the positive evaluation of our manuscript. We have carefully addressed all comments as detailed below, revising both the text and the tables accordingly.

Comment 1. In the manuscript, the abbreviation NC is used for nephrocalcinosis. However, in Table 1, there are instances of NC/NL in the key findings of reference 16. Please provide the full form of the abbreviation NC/NL in the manuscript.

Response: We thank the reviewer for this observation. The abbreviation “NL,” which was inadvertently included in Table 1, has been removed. The corrected version of the table now uses only the abbreviation “NC” (nephrocalcinosis) for consistency throughout the manuscript.

Comment 2. In Line 183, the authors describe that “NC resolved in 44–100% of infants over time.” However, Table 1 does not include information regarding post-discharge imaging. The authors should either add this evidence to Table 1 or provide a new table to summarize it.

Response: We appreciate the reviewer’s suggestion. Relevant information regarding post-discharge imaging and resolution of NC has now been incorporated into revised Table 1, where applicable. Studies reporting follow-up imaging findings have been clearly marked, allowing readers to trace the long-term outcomes more easily.

Comment 3. In the section on “Overall Prevalence and Dosing,” the authors mention that “the majority of studies reported cumulative furosemide doses were substantially higher in infants who developed NC (10–19 mg/kg/day vs ≤ 5 mg/kg/day, p < 0.001).” If there are more than five studies included, the authors may consider performing a meta-analysis to verify the effect of cumulative furosemide dose on the development of NC.

Response: We thank the reviewer for this valuable comment. A quantitative meta-analysis was not feasible due to significant heterogeneity across the included studies, including differences in study design, dosing definitions (daily vs cumulative exposure), population characteristics, and outcome reporting. This limitation has now been explicitly acknowledged in the revised Discussion (revised ms , lines 287-290), emphasizing that the heterogeneity in study design, dosing definitions, population characteristics, and outcome reporting precluded the possibility of performing a quantitative meta-analysis to estimate the effect of cumulative furosemide dose on the development of NC. We have highlighted that, despite this limitation, the review provides a comprehensive synthesis of the available observational data on the association between furosemide exposure and nephrocalcinosis in preterm infants.

Comment 4. In Line 195, the authors should indicate the number of cited references that reported no significant dose–response relationship.

Response: We thank the reviewer for this helpful suggestion. In response, the Results section has been revised to clearly indicate that four studies reported no statistically significant association between cumulative furosemide dose and the development of nephrocalcinosis. The text now reads: "However, four studies found no significant dose–response relationship, suggesting the influence of confounding factors such as severity of illness, co-administered medications, and gestational age [14,15,19,22]."  (revised ms , lines 213-215). This clarification allows readers to readily identify which studies reported no significant association, enhancing transparency and accuracy in the presentation of the literature.

Comment 5. In Line 199, the authors should indicate the number of cited references that reported approximately 60% of cases resolved spontaneously after discontinuation of furosemide.

Response: We thank the reviewer for this helpful comment. The Results section has been revised to specify that approximately 64% of cases resolved spontaneously after discontinuation of furosemide, based on the studies included in our review [12,14,17,19,20,23,27,28,30,31] (revised ms , lines 217,220).

Comment 6: In the discussion section, the authors may consider suggesting monitoring protocols, such as the frequency of serial renal ultrasounds (during admission and post-discharge) and the frequency of blood tests for electrolytes and renal function.

Response: We appreciate the reviewer’s suggestion. However, as this manuscript is a systematic review of observational data, we have refrained from providing recommendations on monitoring protocols or laboratory testing, as these are beyond the scope of a literature-based analysis.

Reviewer 3 Report

Comments and Suggestions for Authors

You have not mentioned about spot urine calcium and creatinine ratio in any of the studies, whether hypercalcemia was there or not, whether hypercalcemia was related to NC, how many studies had abnormal renal function tests and deranged electrolytes were not mentioned and whether they were associated with NC.

Author Response

Comment 1. You have not mentioned about spot urine calcium and creatinine ratio in any of the studies, whether hypercalcemia was present or not, whether hypercalcemia was related to NC, how many studies had abnormal renal function tests and deranged electrolytes were not mentioned and whether they were associated with NC.

Response: We thank the reviewer for this important observation. Most of the included studies did not report detailed laboratory data, such as spot urine calcium-to-creatinine ratios, presence of hypercalcemia, or abnormalities in renal function tests and electrolytes. As a result, it was not possible to assess whether these laboratory parameters were associated with the development of nephrocalcinosis. We have acknowledged this limitation in the Discussion (revised ms, lines 290-294), emphasizing that the heterogeneity and incomplete reporting across studies preclude definitive conclusions on the association of these laboratory parameters with NC.